# Mapping Soil-Transmitted Helminth Parasite Infection in Rwanda: Estimating Endemicity and Identifying At-Risk Populations

**DOI:** 10.3390/tropicalmed4020093

**Published:** 2019-06-14

**Authors:** Eugene Ruberanziza, Kei Owada, Nicholas J. Clark, Irenee Umulisa, Giuseppina Ortu, Warren Lancaster, Tharcisse Munyaneza, Aimable Mbituyumuremyi, Ursin Bayisenge, Alan Fenwick, Ricardo J. Soares Magalhães

**Affiliations:** 1Neglected Tropical Diseases and Other Parasitic Diseases Unit, Malaria and Other Parasitic Diseases Division, Rwanda Biomedical Center, Ministry of Health, Kigali, Rwanda; ruberanzizaeugene@gmail.com (E.R.); umulisa5@gmail.com (I.U.); bayursin@gmail.com (U.B.); 2UQ Spatial Epidemiology Laboratory, School of Veterinary Science, the University of Queensland, Gatton 4343, Queensland, Australia; k.owada@uq.edu.au (K.O.); nicholas.j.clark1214@gmail.com (N.J.C.); 3Children Health and Environment Program, Child Health Research Centre, The University of Queensland, South Brisbane 4101, Queensland, Australia; 4Schistosomiasis Control Initiative (SCI), Department of Infectious Diseases Epidemiology, Imperial College, London SW7 2AZ, UK; giuseppina_ortu@outlook.com (G.O.); a.fenwick@imperial.ac.uk (A.F.); 5The END Fund, New York, NY 10016, USA; wlancaster@end.org; 6Microbiology Unit, National Reference Laboratory (NRL) Division, Rwanda Biomedical Center, Ministry of Health, Kigali, Rwanda; rukabutha@yahoo.com; 7Malaria and Other Parasitic Diseases Division, Rwanda Biomedical Center, Ministry of Health, Kigali, Rwanda; aimable.mbituyumuremyi@rbc.gov.rw

**Keywords:** *Ascaris lumbricoides*, Trichuris trichiura, hookworm, soil-transmitted helminth, spatial epidemiology, Rwanda

## Abstract

Soil-transmitted helminth (STH) infections are globally distributed intestinal parasite infections caused by *Ascaris lumbricoides*, *Trichuris trichiura*, and hookworms (*Ancylostoma duodenale* and *Necator americanus*). STH infection constitutes a major public health threat, with heavy burdens observed in many of the world’s tropical and subtropical regions. Mass drug administration and sanitation improvements can drastically reduce STH prevalence and associated morbidity. However, identifying targeted areas in need of treatment is hampered by a lack of knowledge on geographical and population-level risk factors. In this study, we applied Bayesian geostatistical modelling to data from a national school-based STH infection survey in Rwanda to (1) identify ecological and population-level risk factors and (2) provide comprehensive precision maps of infection burdens. Our results indicated that STH infections were heterogeneously distributed across the country and showed signatures of spatial clustering, though the magnitude of clustering varied among parasites. The highest rates of endemic clustering were attributed to *A. lumbricoides* infection. Concordant infection patterns among the three parasite groups highlighted populations currently most at-risk of morbidity. Population-dense areas in the Western and North-Western regions of Rwanda represent areas that have continued to exhibit high STH burden across two surveys and are likely in need of targeted interventions. Our maps support the need for an updated evaluation of STH endemicity in western Rwanda to evaluate progress in MDA efforts and identify communities that need further local interventions to further reduce morbidity caused by STH infections.

## 1. Introduction

Infection with soil-transmitted helminth parasites (STHs) is a global disease threat [1,2,3,4,5,6]. STH infections are endemic in the world’s tropical and sub-tropical regions, with some of the highest burdens occurring in sub-Saharan Africa [7]. Symptoms include malnutrition, anemia, and in children, poor mental and physical development [8,9]. Global Atlas of Helminth Infection estimates indicate >1.7 billion people were affected by STHs in 2010, placing STH-related morbidity among the world’s most common neglected tropical diseases [7,10]. Reducing this number will rely on strong government support as well as coordinated efforts involving stakeholders such as the pharmaceutical industry, development agencies, and the scientific community [11].

Mass de-worming and targeted sanitation improvements have the power to drastically reduce STH prevalence and associated morbidity [12,13]. Regular delivery of benzimidazole-based anthelminthics to school-aged children is considered the primary mitigation and intervention strategy [14,15]. Yet effective reduction in STH-related morbidity is limited by a poor understanding of geographical patterns in current infection risk and the growing threat of resistance that may stem from untargeted mass drug roll-outs [16,17,18,19,20]. Moreover, prevalence can be temporally variable: improvements in sanitation or living conditions can reduce burdens, while population growth and economic unrest can increase infection rates [7]. Even following effective treatment regimes, continual re-infection is likely in highly contaminated environments [21,22]. Identifying areas in need of targeted STH intervention is thus a major challenge requiring ongoing monitoring efforts.

Evidence-based programs that use high quality infection data and appropriate statistical approaches are essential to provide recommendations for infectious disease control [17,23,24]. These should ideally consider the ecology of the study organisms. The geographical distributions of aetiological agents causing STH infections (*Ascaris lumbricoides*, *Trichuris trichiura*, and hookworms *Ancylostoma duodenale* and *Necator americanus*) are known to be driven by climatic and environmental heterogeneity [25]. For *A. lumbricoides* and *T. trichiura*, eggs remain viable in soil for several months, while hookworm larvae can survive for several weeks. However, survival rates are strongly dependent on environmental conditions [17,26,27]. Geostatistical models that can detect environmental correlates of infection risk have emerged as the tools of choice for delineating regions requiring targeted interventions [4,21,25,28,29].

Rwanda is a landlocked country situated in central Africa. It has a surface area of 26,338 square kilometers and is bordered by Uganda to the North, Tanzania to the East, the Democratic Republic of the Congo (DRC) to the West, and Burundi to the South. The nation is densely populated, has an agriculture-based economy, and a 39% poverty rate [30]. In 2008, a national survey involving over 8000 schoolchildren sampled across 30 districts recorded that 66% of individuals were infected with an STH [22,31]. This placed Rwanda among the most heavily infected STH-endemic regions in the world [7]. Following this report, initial rounds of mass drug administration were rolled out on a national level. Estimates indicate that over 18 million doses of anthelminthics were administered to over four million individuals across the country. This rollout was considered effective, resulting in high cure rates for treated individuals [31]. Yet knowledge of current infection rates and associated risk factors is limited. Social and political unrest within Rwanda (and in neighboring Burundi and DRC) in the 1990s caused widespread disruption to the public health system, leading to a of ratio health providers to the general public that was far below that recommended by the World Health Organization [32]. This resulted in a hampered ability to sustain disease surveillance monitoring [33]. While a return to stability has driven a commendable turnaround in health workforce training and public health outcomes [34], disease monitoring remains a challenge.

Despite some STH research conducted following the 2008 national report [22,31,35], there is a poor understanding of which areas in Rwanda are in most need of interventions. We address this knowledge gap by using geostatistical models to map estimated infection prevalence and identify spatial clusters of high risk for *A. lumbricoides*, *T. trichiura*, and hookworm spp. (*A. duodenale* and *N. americanus*) in Rwandan school children. Although many such geostatistical mapping studies have confirmed that their models generate realistic estimates of their observed data [28,36], it is important to note that these surveys only generate estimates of true prevalence and should be used to formulate guidelines for future study.

## 2. Results

### 2.1. Summary Statistics of STH Prevalence in Rwandan Schoolchildren

Using binomial logistic regressions to analyze STH infections based on a 2014 Kato-Katz survey of 9226 Rwandan schoolchildren, we found that *A. lumbricoides* was the most prevalent STH parasite (observed prevalence = 37%) followed by *T. trichiura* (23%; Table 1). Hookworm prevalence was considerably lower at 5% (Table 1). Geographical variation in observed prevalence for *A. lumbricoides* and *T. trichiura* showed concordant patterns with Northern, Western and the Southwestern parts of country highly affected (Figure 1: Panels A and B). Hookworm prevalence was low across the country (Figure 1: Panel C).

### 2.2. Spatial Clustering in STH Prevalence and Risk Prediction

Semivariograms based on residuals from logistic regressions demonstrated a tendency for spatial clustering for all three parasites, though this pattern was strongest for *A. lumbricoides* followed by hookworm (Table 2; See Appendix A for visual representations of semivariograms). Using geostatistical models to quantify possible predictors while accounting for spatial effects, we found that the Normalized Difference Water Index (NDWI) was a positive predictor of infection probability for all three parasite groups (Table 3). In fact, this effect was the only significant predictor of hookworm infection. In contrast, we also found an effect of participant age on *T. trichiura* infection probability, with individuals aged between 12 to 14 years at higher risk compared to those aged between 9 to 11 years, and positive associations between both the Normalized Difference Vegetation Index (NDVI) and NDWI for *A. lumbricoides* infection probability (Table 3). Estimated radii of spatial clusters for hookworm was larger compared to those of *A. lumbricoides* and *T. trichiura* (Table 3). No substantial difference in prevalence between the sexes was evident for any of the three parasites (Table 3).

Spatial predictions of infection risk for boys aged between 12 and 14 years, the group with the highest overall burden of infection in our dataset, identified regions with highest current need of targeted interventions. Predicted probability of *A. lumbricoides* infection ranged between 40 to 50% across the Northern, Western, and Southwestern provinces of Rwanda (Figure 2: Panel A). Districts with the highest predicted probability of infection (>50%) for each of these provinces were, for the Northern province: Musanze, Gakenke, Burera, Rulindo, and Gicumbi districts; for the Western province: Rubavu, Nyabihu, Rutsiro, Ngororero, Karongi, Nyamasheke, and Rusizi districts; and for the Southwestern Province: Nyamagabe and Nyaruguru districts (Figure 2: Panel A; see Appendix A for a map of Rwanda’s current administrative districts). Similarly, predicted prevalence of *T. trichiura* infection ranged between 20 to 40% across Northern and the Western provinces, with predicted prevalence above 50% observed in the Burera and Musanze districts in the Northern province and the Karongi, Ngororero, Nyabihu, Nyamasheke, Rubavu, Rusizi, and Rutsiro districts in the Western province (Figure 2: Panel B; Appendix A). Predicted prevalence of hookworm infection was generally less than 10% across Rwanda, with the highest predicted prevalence observed in a small localized area in the Ngoma district in the Eastern province (30–40%; Figure 2: Panel C; Appendix A).

### 2.3. Estimated Current Burdens of A. lumbricoides, T. trichiura and Hookworms

Extrapolating infection probability estimates to a derived 2018 population raster map revealed that approximately 589,673 (95% CI: 575,836, 605,582) individuals aged 12 to 14 years are currently infected with *A. lumbricoides* in Rwanda. Approximately 332,144 (95% CI: 324,031, 349,579) and 83,749 individuals (95% CI: 78,761, 93,031) in this at-risk group are estimated to be infected with *T. trichiura* and hookworm, respectively (Table 4).

We identified target areas where the likelihood of *A. lumbricoides, T. trichiura* and hookworm burdens among individuals aged 12 to 14 years are at their highest (Figure 3: Panel A to C). For *A. lumbricoides*, the Musanze district (Muhoza sector), Rubavu district (Rubavu, Gisenyi, Nyamyumba, and Nyakiriba sectors), Rusizi district (Nkombo and Mururu sectors), and Nyarugenge district (Gitega and Kimisagara sectors) had the highest number of individuals with predicted *A. lumbricoides* infection, with some areas exceeding 534 infected people per square kilometre (Figure 3: Panel A; Appendix A). The distribution of heavy *T. trichiura* burdens, where up to 654 people per square kilometre were predicted to be infected, was more localized around Gisenyi, Rugerero, and Nyamyumba sectors in Rubavu district in the Western province, the border of the Democratic Republic of the Congo, Mururu sector in Rusizi district, and Nkombo sector, which is located around the southern end of Lake Kivu in the Southwestern province (Figure 3: Panel B; Appendix A). Predicted prevalence of hookworm shows a wide distribution across Rwanda, though estimated burdens are far lower than for the other two parasites (with some areas exceeding 90 people per square kilometer; Figure 3: Panel C). These burdens are predicted to occur in localized areas of the Eastern province including Kayonza district (Mukarange sector), Rwamagana district (Kigabiro sector), Ngoma district (Kibungo, Mugesera, Remera, Rukumberi, and Zaza sectors), and Nyagatare district (Nyagatare and Rukomo sectors); the Northern province including Burera district (Rugengabari sector); the central Kigali area including Nyarugenge district (Gitega, Kimisagara, Nyakabanda, and Rwezamenyo sectors); and the Western province including Ngororero district (Ngororero sector) and Rusizi district (Gihundwe, Kamembe, Muganza, Ngororero, and Nkombo sectors; Appendix A).

### 2.4. District-Level Temporal Trends in Soil-Transmitted Helminth Endemicity

Visualizing the proportions of surveyed individuals infected with at least one STH parasite across two surveys (2008 and 2014; aggregated by year and by district), revealed a reduction in the number of districts with high burden estimates over time (Figure 4 and Appendix A). In 2008, district-level prevalence estimates were above 50% for the majority of the country, with districts in the Northern, Western, and Southwestern regions generally showing the highest burdens. Several districts in the Eastern and Southern regions were also estimated to have moderate (50–70% infection prevalence) or high (≥70%) burdens in 2008, however, many of these showed considerable decreases in the 2014 survey (Figure 4). Of highest concern are districts in the Western and North-Western regions, which continue to exhibit high burdens (Figure 4).

## 3. Discussion

We have provided comprehensive national precision maps of STH infections in Rwanda. Our findings that STH infections are heterogeneously distributed and show signatures of spatial clustering broadly agree with mounting evidence from other endemic regions [4,16,28]. Concordant infection patterns among the three parasite groups highlighted populations at most risk. Population-dense districts in the Western and Northwestern regions of Rwanda represented areas of high STH burden in 2008 and continue to do so, justifying the need for targeted interventions in these regions. Our high-resolution maps provide useful insights that can be used for tailoring monitoring programs. However, we stress that further data will be necessary to refine estimates as treatment programs are rolled out, supporting the argument that continuously updated geostatistical models are essential for informing STH mitigation [21,25,28,37].

### 3.1. Delineating Regions in Rwanda at Greatest Need of STH Mitigation

Rwanda is one of the most densely populated countries in Africa and along with malaria infection and dietary iron intake, infection by intestinal parasites is one of the most common causes of anemia [38,39]. While blood-feeding hookworms may be considered the primary cause of parasite-driven anemia, there are numerous case reports of iron-deficiency anemia resulting from infection by other STH parasites [40,41]. In Rwanda, up to 37% of children aged 6–59 months are estimated to be living with some level of anemia. Our modelling indicates districts in the Western and North-Western regions of the country continue to harbor high STH burdens following initial mass drug rollouts and should be considered priority areas for treatment and subsequent monitoring. Generally consisting of mountain highlands, the Virunga volcano range, and the foothills of Lake Kivu [38], we estimate that prevalence of *A. lumbricoides* and *T. trichiura* reaches over 50% among the at-risk population in some of these districts. These areas also tend to show high endemicity spatial clustering, particularly for *A. lumbricoides* infection. Predicted prevalence of hookworm showed a wide distribution across Rwanda, and our results indicated that estimated burdens are much lower than for *A. lumbricoides* and *T. trichiura*. This could be explained by consistent evidence that adults are more at risk of hookworm infection compared to school-aged and younger population [42] and the challenges associated with the detection of hookworm eggs using the Kato-Katz technique [43]. Collectively, our results suggest local factors in Rwanda’s Western and North-Western regions are highly conducive to STH transmission. Population density likely plays a strong role, as these regions are among the most heavily populated in the country [30,44]. STH, along with many other parasitic worms, can maintain high transmission rates in regions with overcrowding combined with poor level of sanitation [45,46]. As such, the level of urbanization a region exhibits has been highlighted as a key indicator of potential parasite spread [47,48,49,50]. In Rwanda, differences between rural and urban areas in STH infection and post-treatment re-infection rates have been observed and are thought to possibly reflect differences in education levels, poverty, and availability of WASH [22]. It stands to reason then that the lack of sanitation facilities may be an important contributor to STH transmission in these population-dense areas. Indeed, malnutrition and diarrhea are common causes of hospital-based mortality among children in Rwanda [34], suggesting that lack of adequate sanitation is a national public health challenge. While we did not directly investigate potential influences of urbanization or sanitation levels on STH infection rates, this represents a tractable aim that should be considered in future studies in Western Rwanda.

### 3.2. Influence of Ecological Variables on STH Infection Probability

In addition to delineating regions in need of intervention, a key aim of infectious disease modelling is to uncover ecological drivers of observed infection patterns [51,52,53,54]. Capabilities to uncover ecological correlates of pathogen distributions and infection rates have greatly improved with the widespread availability of satellite imagery [16,25,51,55]. Here, we identified NDWI as an important predictor of infection probability for all three STH parasite groups. This is perhaps unsurprising given the well-established importance of environmental conditions, particularly those that influence soil moisture, on STH transmission [25,56]. Infectious stages of these parasites thrive in warm and moist soils, which is a key indicator of their strongly tropical and subtropical global distributions [57]. Our findings indicate that regional differences in infection rates may also be driven by this environmental property.

Interestingly, NDVI was only correlated with infection probability for one of the three parasite groups (*A. lumbricoides*), despite vegetation indices being consistent predictors of infection risk for *T. trichiura* in other regions [16,28]. Indeed, evidence indicates that low NDVI values correlate with increased survival and transmission rates for both of these parasite species [17]. Though the 95% confidence interval of our NDVI estimate for *T. trichiura* only marginally overlapped with zero, why we did not find evidence of a stronger association requires further exploration that considers the biology of the organism. Considering the long lifespans of adult *T. trichiura* worms (1–2 years) and the relatively high fecundities of intestinal heminths in general [57], we may expect overall transmission rates to depend more on survival of adults in the human host than on environmental conditions. When mass rollouts of preventative drugs are infrequent adult worms would be expected to live relatively long lifespans compared to regions where drug delivery is more frequent. If this were the case, then egg production may be so great that transmission can occur even when soils are relatively harsh for egg survival [58,59]. This could partially explain why NDVI was less important for predicting *T. trichiura* infection rates in Rwanda, though further additional empirical evidence is clearly needed to test this idea.

### 3.3. Study Limitations

Our findings shed new light on the spatial distributions of STH infections in Rwanda. However, there are certain limitations that should be addressed. Uncertainties in remote-sensed variables are commonly ignored when producing raster maps and the data we used is no exception [17,29]. In addition, our prediction maps used at-risk population estimates that come with their own inherent uncertainties. We relied on data from the United Nations (UN) population database to generate our estimates of the number of children at risk. These may be overestimated, particularly because data on the proportion of persons within this age group is available only for children aged 10 to 14 years. Therefore, our total population map for the at-risk group was created by multiplying prediction maps for boys aged 12 to 14 years old by a population density map of boys aged 10 to 14 years. Moreover, these estimates were calculated using the UNDP average annual rate of population change, which may not reflect spatio-temporal variation in population changes. Estimates of infection rates in adults should also be considered when attempting to gain accurate insights into population-level risk factors [15]. Parasite egg counts in our study were based on a single sample, which limits the Kato-Katz test performance for detecting light infection intensities [60,61]. We also did not adjust for potential measurement error in eggs per gram of feces (epg) counts. While this may not affect our spatial or spatiotemporal inferences, including data on egg burdens might be one way to improve precision of estimates for associations between predictors and STH burden. Finally, we did not assess whether infection with one STH parasite changes an individual’s risk of infection with other parasites in this study. Co-infections between parasite species are ubiquitous in nature, and empirical evidence is mounting to suggest that parasite interactions can influence infection risk and/or disease manifestations [4,53,62,63].

### 3.4. Conclusions and Future Directions

Nation-wide infection rates among Rwandan schoolchildren ranged from 22–38% for *A. lumbricoides* and *T. trichiura*. While these estimates are lower than 2008 estimates, they constitute a serious public health concern, especially considering the impacts that STH infections have on childhood growth and development [3,8,9,64]. Initiatives to rollout mass deworming programs and reduce STH burdens in endemic regions have received overwhelming global support [11]. With this support comes not only an incredible increase in available resources, but a renewed necessity to broaden our understanding of STH geographical distributions and population-level risk factors. A primary impediment to the development and implementation of cost-effective STH control is the current lack of geographical distribution studies in many endemic regions [17,57]. Studies such as ours can address this gap.

Unfortunately, STH infections are likely to remain a substantial threat to public health while poverty persists in developing tropical and subtropical nations [57]. When soils are particularly conducive to parasite survival/transmission, as they seem to be in the Northern and Western regions in Rwanda, mitigation must rely on multi-pronged strategies to (1) treat infected individuals as a means of reducing egg production; (2) improve local sanitation measures to reduce soil contamination; and (3) increase awareness and education standards [13,14,65]. Our comprehensive national precision maps of STH infection rates can be used as a tool to tailor these mitigation strategies in Rwanda. In this way, we hope that our study helps to fuel the already commendable efforts to reduce STH-related morbidity in the world’s tropical and subtropical regions.

## 4. Materials and Methods

### 4.1. Ethics Statement

Ethical clearance for this analytical study was provided by the National Ethics Committee in Rwanda (Sep 2014, Ref No: 261/RNEC/2014).

### 4.2. STH Infection Data

We used data from a national school-based survey aimed at detecting the presence of *A. lumbricoides*, *T. trichiura* and hookworm infections in Rwanda. The survey was administered across 186 schools between June and mid-July 2014. The mapping unit considered for STH was the district. Adjustments were made to ensure at least 5 schools in each district for STH mapping and the 31 proposed sentinel schools were included- 12 former and 19 new sentinel schools. Fifty children of the required age (primarily 13–14 years but also 10–16 years to reach 50) were randomly selected in each selected school [66]. For each individual participant involved in the survey, binary presence-absence of parasite infection based on the presence of at least one egg was assessed using the widely-implemented Kato-Katz method, which involves microscopic examination of duplicate 41.7 mg fecal smears taken from participants [67].

### 4.3. Extraction of Environmental Covariate Information

Environmental variables were extracted at the school level from remote-sensing data warehouses to be used as covariates in predictive models. We extracted information for three variables known to reflect heterogeneity in STH survival and transmission rates. These were: land surface temperature (LST), extracted from WorldClim), and NDVI and NDWI, both of which were extracted from the National Oceanographic and Atmospheric Administrations’ Satellite and Information Services (NOAASIS). LST is appropriate because soil-transmitted helminths exhibit thermal survival thresholds [68]. NDVI and NDWI variables capture moisture levels of the soil, which is also known to impact survival of infective stages [2,17,21]. All variables were obtained at a 1km x 1km grid cell resolution using the Google Earth Engine in ArcGIS version 10.4.0.5524 [69] and were standardized to unit variance prior to analysis.

### 4.4. Statistical Analysis

A total of 9251 observations were available in the original dataset. However, 25 observations were missing age information and one observation was missing *T. trichiura* and hookworm infection status information. These observations were excluded from analysis. The final dataset included a total of 9226 (for *A. lumbricoides* infections) and 9225 (for *T. trichiura* and Hookworm infections) participants from 186 schools. Ages of participants were stratified into a three-level categorical variable (9 to 11 years old; 12 to 14; and 15 to 18 years old). The average age of the 9226 individuals included in our analysis was 13.31 years. School centroids were estimated using a shapefile representing the current administrative units of Rwanda, i.e., sectors. The shapefile was obtained from the geographic data warehouse DIVA GIS and centroids were delineated using the geographical information system (GIS) software Quantum GIS version 1.7.3 [70]. Infection data and environmental data were then linked to school centroids.

#### 4.4.1. Frequentist Models for Variable Selection and Examination of Spatial Autocorrelation

We used maximum likelihood logistic regression models (binomial errors with logit link function) to test for possible spatial autocorrelation in STH infection probability. Regression predictors included participant sex and age (categorical variables) and the scaled the environmental predictors LST, NDVI, and NDWI. Residuals for the final models were extracted and examined for possible spatial autocorrelation by generating semivariograms using functions in the ‘geoR’ package [71] in R version 3.1.1 (The R foundation for statistical computing, Vienna, Austria). A semivariogram is a graphical representation of the spatial variation in a dataset; residual semivariograms represent the spatial variation left unexplained after inclusion of covariates in a model. The semivariogram is characterised by three parameters: the partial sill, which is the spatially structured component of the semivariance (indicative of the tendency for geographical clustering), the nugget, which is the spatially unstructured component of the semivariance (representing random variation, very small-scale spatial variability, or measurement error), and the range, which is the distance at which locations can be considered independent (indicative of the average size of geographical clusters). The tendency for geographical clustering within a region (i.e., proportion of variation that is due to spatial proximity) was estimated by dividing the partial sill by the sum of the nugget and the partial sill [23].

#### 4.4.2. Geostatistical Models of Soil-Transmitted Helminth Infection Risk

Because analysis of spatial autocorrelation revealed some level of spatial clustering for all parasites (see Results above), we implemented geostatistical models to simultaneously account for our specified covariates and to explore this autocorrelation. We assumed the presence-absence data vectors for each parasite were random draws from an infection probability according to a Bernoulli distribution. We modelled this probability using a logit-link transformation of a linear parameterization that included an intercept, the specified fixed effects and a multivariate normal covariance matrix capturing distances between pairs of sampling points as the geostatistical random effect (using the *spatial.exp* function BUGS language, which is essentially a kriging model). We fitted spatial models in a Bayesian framework using Markov Chain Monte Carlo (MCMC) sampling based on the Gibbs sampler in the open-source software OpenBUGS [72]. We specified vague normal priors with mean = 0 and variance = 100 (i.e., precision = 0.01) for intercepts and regression coefficients. The geostatistical random effects were assumed to follow a normal distribution, with mean = 0 and variance = 1/*tau*, where the precision *tau* was given a gamma prior distribution (shape = 0.001, scale = 0.001). For each model, a burn-in of 5000 MCMC iterations was used, followed by 10,000 iterations during which values for the intercept and regression coefficients were stored for parameter estimation and generation of predictive maps. Diagnostic tests for convergence were assessed visually based on inspection of posterior density and trace plots. Significance of effects was inferred based on whether 95% confidence intervals of posterior estimates did not overlap with zero. Separate regressions were run for each parasite group.

Model predictions were used to generate representative maps of parasite prevalence across Rwanda for boys aged between 12 to 14 years, the subgroup with the highest prevalence of infections in our dataset. Predictions were made at the nodes of a 0.03 × 0.03 decimal degree grid (approximately 3 km^2^). The mean and standard deviation were extracted from the posterior distributions of predicted risk. We generated marginal predictions using the *spatial.unipred* command in OpenBUGS, which carries out single site predictions to yield marginal prediction intervals for each study location.

#### 4.4.3. Estimating Current Burdens of *A. lumbricoides*, *T. trichiura*, and Hookworm Infections in an At-Risk Population in Rwanda

We chose to demonstrate geographical discrepancies in *A. lumbricoides*, *T. trichiura*, and hookworm endemicity and infection risk for boys aged between 12 to 14 years old. We multiplied a raster map of the estimated total population size of this select group (estimated for the year 2018) by the predicted prevalence of *A. lumbricoides*, *T. trichiura*, and hookworm infections to derive a map of the total number of children at risk in each grid cell. To derive the 2018 population raster map, we retrieved a 2015 population density raster from the Center for International Earth Science Information Network (CIESIN) [44]. The CIESIN uses National Institute of Statistics Rwanda’s Fourth Population and Housing Census 2012 data to estimate their population raster map. The 2015 raster was then multiplied by the reported UNDP average annual rate of population change between 2015 to 2020 (i.e., 2.53%). This map was then multiplied by the proportion of 12 to14 year old boys (i.e., 12.2%) in order to produce a raster map of the estimated total number of children belonging to this focal group in 2018. All spatial calculations and plots were conducted in ArcGIS version 10.4.0.5524 [69].

#### 4.4.4. Visualising Temporal Trends in STH Endemicity

Analyses above were conducted at the site level to produce up-to-date and high-resolution estimates of geographical variation in STH burdens. To investigate possible temporal trends in spatial patterns, we supplemented site-level analyses using STH survey datasets from 2008 and 2010 that were only available as aggregated counts (total participants surveyed and total infected) at the district level. The 2014 data was aggregated in a similar fashion, and we produced chloropleth maps of estimated endemicity for each time point (2008, 2010, and 2014).

## Figures and Tables

**Figure 1 tropicalmed-04-00093-f001:**
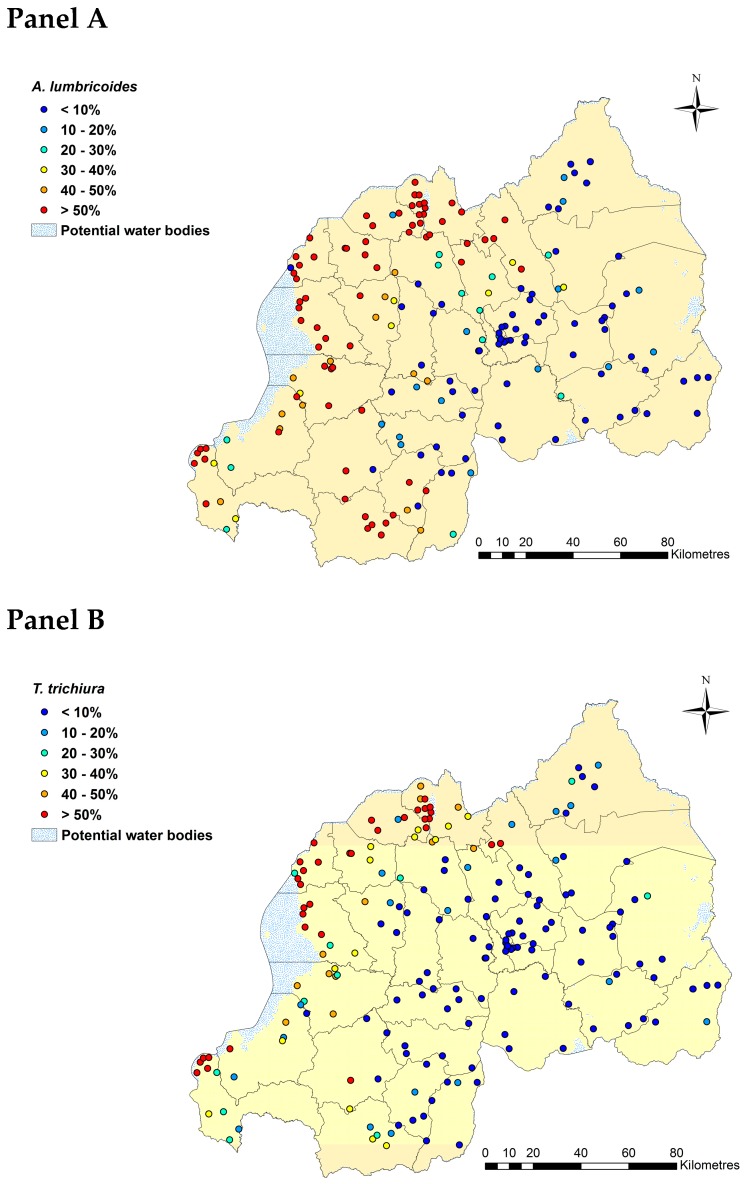
Observed prevalence of *A. lumbricoides* (**Panel A**), *T. trichiura* (**Panel B**), and hookworm (**Panel C**) based on Kato-Katz in Rwanda in 2014. This figure was produced in ArcMap 10.4 (ESRI, Redlands, CA) using a shapefile representing Rwanda’s current administrative units (obtained from the data warehouse DIVA GIS).

**Figure 2 tropicalmed-04-00093-f002:**
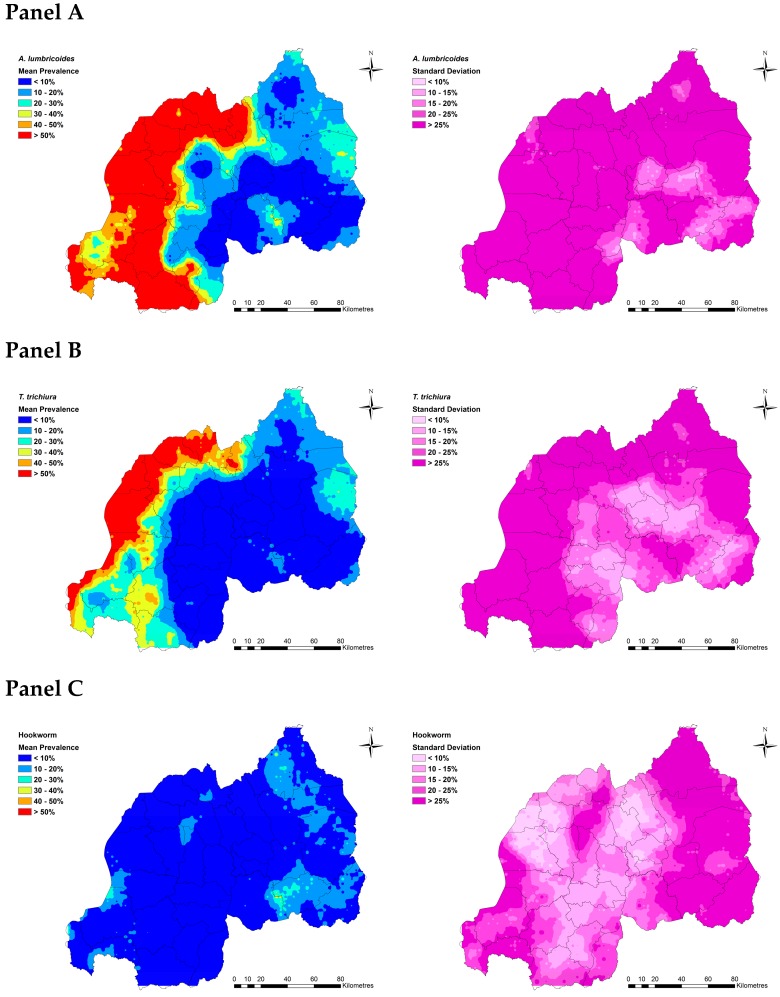
Predicted prevalence of *A. lumbricoides* (**Panel A**), *T. trichiura* (**Panel B**), and hookworm (**Panel C**) for boys aged 12–14 years in Rwanda in 2018. Total population raster map is based on population density grid estimated using National Institute of Statistics Rwanda, Fourth Population and Housing Census 2012 data. This figure was produced in ArcMap 10.4 (ESRI, Redlands, CA) using a shapefile representing Rwanda’s current administrative units (obtained from the data warehouse DIVA GIS).

**Figure 3 tropicalmed-04-00093-f003:**
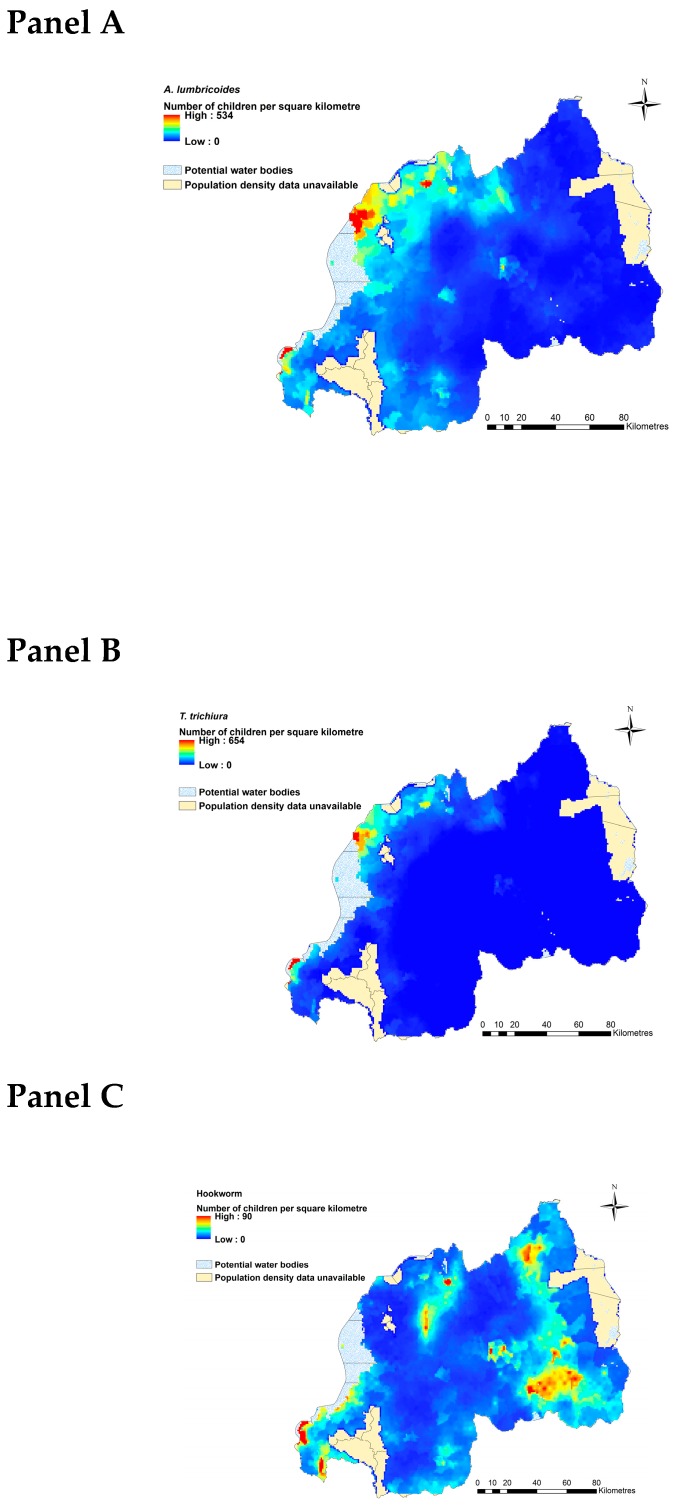
Distribution of number of children with predicted to harbour *A. lumbricoides* infection (**Panel A**), *T. trichiura* infection (**Panel B**), and hookworm (**Panel C**), (people per square kilometre) in Rwanda in 2018. Total population raster map is based on population density grid estimated using National Institute of Statistics Rwanda, Fourth Population and Housing Census 2012 data. This figure was produced in ArcMap 10.4 (ESRI, Redlands, CA) using a shapefile representing Rwanda’s current administrative units (obtained from the data warehouse DIVA GIS).

**Figure 4 tropicalmed-04-00093-f004:**
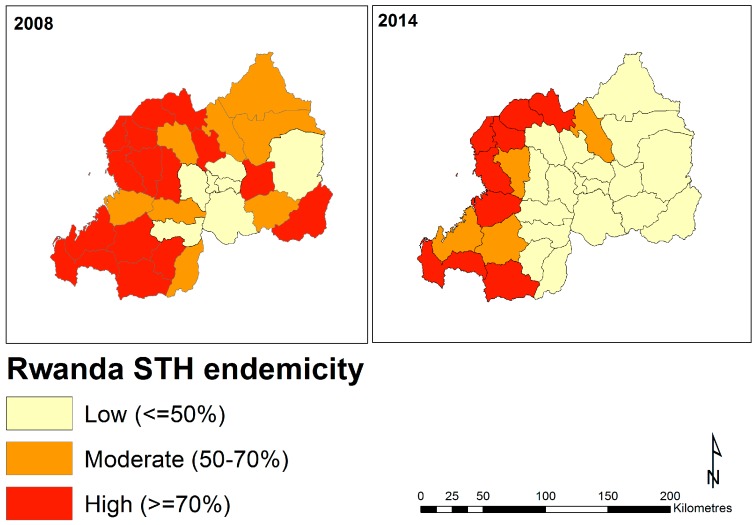
Chloropleth maps of estimated soil-transmitted (STH) helminth endemicity, represented as proportions of surveyed individuals found to be infected with at least one STH parasite (*A. lumbricoides*, *T. trichiura*, and hookworm), in Rwanda across two survey periods. This figure was produced in ArcMap 10.4 (ESRI, Redlands, CA) using a shapefile representing Rwanda’s current administrative units (obtained from the data warehouse DIVA GIS).

**Table 1 tropicalmed-04-00093-t001:** Summary statistics of participant demographics and prevalence of soil-transmitted helminth infections across schools in Rwanda in 2014. CI; credible interval; LST: land surface temperature; NDVI: Normalized Difference Vegetation Index; NDWI: Normalized Difference Water Index.

Variable	Observation	Mean	95%CI	Missing Data
Number of schools	186	-	-	-
Units (district)	30	-	-	-
Number of individuals surveyed	9251			
**Demographic**				
**Age**	9226	13.31 (Range 9 to 18)	13.29, 13.33	Location 232 R1 to R25 (N = 25)
**Age category**	9226	-	-	-
9 to 11 years old	151 (1.64%)	-	-	-
12 to 14 years old	8931 (96.80%)	-	-	-
15 to 18 years old	144 (1.56%)	-	-	-
***Sex***	9251	-	-	-
Female	4617 (49.9%)	-	-	-
Male	4634 (50.1%)	-	-	-
**Prevalence of infections**				
*A. lumbricoides*	9251	0.37	0.36, 0.38	
*T. trichiura*	9250	0.23	0.22, 0.24	Location 104/R46
Hookworm	9250	0.05	0.04, 0.05	Location 104/R46
**Environmental variables**				
LST 2014	9251	−0.001	−0.021, 0.020	-
NDVI 2014	9251	0.002	−0.018, 0.023	-
NDWI 2014	9251	0.002	−0.019, 0.022	-

**Table 2 tropicalmed-04-00093-t002:** Summary parameters of soil-transmitted helminth spatial semivariograms.

	Observed	Residual
***A. lumbricoides***		
Partial sill	0.167	0.013
Nugget	0.001	0.000
Practical range *	1.686	0.421
% of the variance due to clustering	99.84	99.41
***T. trichiura***		
Partial sill	0.083	0.01
Nugget	0.007	0.005
Practical range *	1.526	0.94
% of the variance that is due to cluster **	92.67	64.76
**Hookworm**		
Partial sill	0.003	0.001
Nugget	0.000	0.006
Practical range *	0.463	0.149
% of the variance that is due to cluster **	88.23	11.07

* Calculation based on practical range multiplied by 111; 1 decimal degree = 111 km; 0.1 = 11 km; 0.01 = 1km; 0.05 = 5 km; 0.005 = 500 m; ** Calculation based on partial sill divided by sill (partial sill + nugget), multiplied by 100.

**Table 3 tropicalmed-04-00093-t003:** Effects from geostatistical models estimating prevalence of *A. lumbricodes*, *T. trichiura* and hookworm in Rwanda in 2014. **Bolded** text displays statistically significant associations.

Coefficient	*A. lumbricoides* Mean (95% CI) ^b^	*T. trichiura* Mean (95% CI) ^b^	Hookworm Mean (95% CI) ^b^
Intercept	−1.651(−3.981, 0.214)	−2.891 (−4.417, −1.121)	−4.585 (−7.184, −2.656)
Male (versus female)	−0.032 (−0.150, 0.086)	0.036 (−0.098, 0.167)	−0.043 (−0.257, 0.159)
Age 12 to 14 years old (versus 9 to 11 years old)	0.357 (−0.502, 1.248)	**0.998 (0.015, 2.024)**	0.996 (−0.890, 3.573)
15 to 18 years old (versus 9 to 11 years old)	0.316 (−0.651, 1.323)	0.791(−0.308, 1.931)	0.542 (−1.703, 3.192)
Normalized difference vegetation index ^a^	**0.166 (0.015, 0.322)**	0.083 (−0.088, 0.249)	−0.117 (−0.337, 0.089)
Land surface temperature ^a^	−0.080 (−0.251, 0.100)	0.045 (−0.141, 0.230)	−0.065 (−0.288, 0.178)
Normalized difference water index ^a^	**0.296 (0.107, 0.491)**	**0.263 (0.074, 0.450)**	**0.308 (0.054, 0.583)**
φ Phi (rate of decay) ^c^	3.154 (0.988, 5.696)	2.912 (0.541, 5.520)	7.911 (3.343, 13.610)
σ^2^Sigma (variance)	5.888 (2.788, 13.550)	6.639 (2.818, 23.750)	2.268 (1.336, 3.910)
Tau (precision)	0.210 (0.074, 0.359)	0.197 (0.042, 0.355)	0.475 (0.256, 0.748)

Results based on Bernoulli Bayesian Geostatistical Model (Burn-in: 5000. Sample: 10,000); ^a^ Variables were standardized to have mean = 0 and standard deviation = 1; ^b^ CI = Credible Interval; ^c^ Measured in decimal degrees; 3/phi determines cluster size; one decimal degree is approximately 111 km at the Equator (the size of the radii of the clusters).

**Table 4 tropicalmed-04-00093-t004:** Predicted number of individuals aged 12 to 14 years with *A. lumbricoides*, *T. trichiura*, and hookworm infection, in Rwanda, 2018.

Total Population for 2015 (in Thousands) ^a^	Annual Population Growth Rate for 2015–2020 (Percentage) ^a^	Percentage of Individuals Aged 12–14y ^a^	Predicted Number of Individuals with Infection in 2018		
			Infections		
			*A. lumbricoides*	*T. trichiua*	Hookworm
11,629.6	2.53	12.20%	589,673	332,144	83,749

Source: **^a^** World Population Prospects 2017 Revision Population Database: Rwanda (Population Division of the Department of Economic & Social Affairs of the United Nations Secretariat, 2017).

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
