# Peer review of "Mapping Soil-Transmitted Helminth Parasite Infection in Rwanda: Estimating Endemicity and Identifying At-Risk Populations"

_tropicalmed, 2019, doi:10.3390/tropicalmed4020093_

Round 1
Reviewer 1 Report
Dear Authors
your paper is interesting for scientific community and as support to action plan of STH infections, and English form is fine. Anyway, the reading process is heavy and non-flowing, probabily because some concepts are often repeated and not well schematized.
Moreover, data in Fig 5 are too much analytical and better represented by the Fig 4 with the chloropleth map. Consequently, you can consider that this figure may be removed or synthesized in some way.
Author Response
Reviewer #1:
Comment #1:
Your paper is interesting for scientific community and as support to action plan of STH infections, and English form is fine. Anyway, the reading process is heavy and non-flowing, probabily because some concepts are often repeated and not well schematized.
Response: Thank you for positive comments on our manuscript and its relevance to action plans. Based on both of the reviewers’ comments, we have removed a redundant table (previously Table 5) from the main text, cleaned up definitions throughout the manuscript and added lines to improve clarity of our findings and interpretations. We believe these changes have improved the layout of the text and we thank the reviewers for their inputs.
Comment #2:
Moreover, data in Fig 5 are too much analytical and better represented by the Fig 4 with the chloropleth map. Consequently, you can consider that this figure may be removed or synthesized in some way.
Response: There is no Figure 5 in our manuscript. We presume the reviewer referring to Table 5, which we agree is a bit heavy for a main text table. We have instead moved this table to the Supplementary Material and changed its label to Table S1.

Reviewer 2 Report
Line 53: Presumably government support. Only need to look at China and reduction of schistosomiasis there in contrast to the Philippines to see the importance of having governments invested in outcomes
Line 93: How similar does this end up to the actual prevalence? Has it been used previously where helminth surveys have been conducted to confirm mapping/model predicted prevalence?
Line 112: STH
Line 121: Should be NDWI? – come back as I now see it is vegetation index after reading the discussion – should be defined here as this is the first mention in the manuscript text
Figure 2 caption should be below figure, ditto figure 3 and figure 4 (which looks to be formatted for figure caption whereas 2 and 3 do not)
Table 3: Are some of the numbers purposefully in bold? Perhaps mention what it means if in bold in the caption for the table
Line 230: STH defined earlier
Line 242: Only hookworm presumably as the other GIT worms are not blood feeders, and prev was down for hookworm
Line 274: previously defined
Line 328: STH
Line 351: Kato-Katz has been used in full through rest of manuscript, either use abbreviation everywhere or not at all
Did not have access to supplementary materials so cannot comment on those.
Author Response
Reviewer #2:
Comment #1:
Line 53: Presumably government support. Only need to look at China and reduction of schistosomiasis there in contrast to the Philippines to see the importance of having governments invested in outcomes.
Response: Thank you for this suggestion. We have changed the line in question to “Reducing this number will rely on strong government support as well as coordinated efforts involving stakeholders such as the pharmaceutical industry, development agencies and the scientific community [11].”
Comment #2:
Line 93: How similar does this end up to the actual prevalence? Has it been used previously where helminth surveys have been conducted to confirm mapping/model predicted prevalence?
Response: This is a good point. Although many such geostatistical mapping studies have confirmed that their models generate realistic estimates of their observed data (see: Soares Magalhaes et al 2015 and Pullan et al 2011 as examples), these surveys will always only generate estimates of true prevalence as the entire cohort of individuals cannot be surveyed. We have changed the line in question to recognise this limitation, it now reads: “We address this knowledge gap by using geostatistical models to map estimated infection prevalence and identify spatial clusters of high risk for A. lumbricoides, T. trichiura and hookworm spp. (A. duodenale and N. americanus) in Rwandan school children. Although many such geostatistical mapping studies have confirmed that their models generate realistic estimates of their observed data [see for example 28,36], it is important to note that these surveys only generate estimates of true prevalence and should be used to formulate guidelines for future study.”
Soares Magalhães, R.J., Salamat, M.S., Leonardo, L., Gray, D.J., Carabin, H., Halton, K., McManus, D.P., Williams, G.M., Rivera, P., Saniel, O., Hernandez, L., Yakob, L., McGarvey, S.T. & Clements, A.C.A. (2015) Mapping the risk of soil-transmitted helminthic infections in the Philippines. PLoS Neglected Tropical Diseases, 9, e0003915.
Pullan, R.L., Gething, P.W., Smith, J.L., Mwandawiro, C.S., Sturrock, H.J., Gitonga, C.W., Hay, S.I. & Brooker, S. (2011) Spatial modelling of soil-transmitted helminth infections in Kenya: a disease control planning tool. PLoS Neglected Tropical Diseases, 5, e958.
Comment #3:
Line 112: STH
Response: Changed
Comment #4:
Line 121: Should be NDWI? – come back as I now see it is vegetation index after reading the discussion – should be defined here as this is the first mention in the manuscript text
Response: Thank you for pointing this out. In fact, both NDWI and NDVI were found to be positively associated with A. lumbricoides infection risk. We have changed this line to now read: “and positive associations between both the Normalized Difference Vegetation Index (NDVI) and NDWI for A. lumbricoides infection probability (Table 3).” Please note that NDWI is defined two lines above this, and so it does not need to be defined here.
Comment #5:
Figure 2 caption should be below figure, ditto figure 3 and figure 4 (which looks to be formatted for figure caption whereas 2 and 3 do not)
Response: We have moved the figure captions below the figures as requested
Comment #6:
Table 3: Are some of the numbers purposefully in bold? Perhaps mention what it means if in bold in the caption for the table
Response: This is correct. We have added a line to the caption that reads: “Bolded text displays statistically significant associations.”
Comment #7:
Line 230: STH defined earlier
Response: We have changed this line to now read: “We have provided comprehensive national precision maps of STH infections in Rwanda.”
Comment #8:
Line 242: Only hookworm presumably as the other GIT worms are not blood feeders, and prev was down for hookworm
Response: In fact, there are numerous case reports of Fe-deficiency caused by other STH infections, which ultimately leads to anaemia (see García-Leiva et al 2008 and Amuga et al 2006 as examples). We have changed the line in question to now read: “While blood-feeding hookworms may be considered the primary cause of parasite-driven anaemia, there are numerous case reports of iron-deficiency anaemia resulting from infection by other STH parasites [40,41].”
García-Leiva, J., Barreto-Zuñiga, R., Estradas, J. & Torre, A. (2008) Ascaris lumbricoides and iron deficiency anemia. The American Journal of Gastroenterology, 103, 1051.
Amuga, G., Onwuliri, C. & Oniye, S. (2006) Relative contribution of hookworm and Ascaris lumbricoides to iron deficiency anemia among school pupils in Nasarawa area, Nigeria. International Journal of Natural and Applied Sciences, 2, 205-209.
Comment #9:
Line 274: previously defined
Response: We have changed this line to now read: “Here, we identified NDWI as an important predictor of infection probability for all three STH parasite groups”
Comment #10:
Line 328: STH
Response: Changed
Comment #11:
Line 351: Kato-Katz has been used in full through rest of manuscript, either use abbreviation everywhere or not at all
Response: We have removed the definition of the acronym here, opting to use the full name throughout. This line now reads: “binary presence-absence of parasite infection based on the presence of at least one egg was assessed using the widely-implemented Kato-Katz method, which involves microscopic examination of duplicate 41.7mg faecal smears taken from participants [67].”
Comment #12:
Did not have access to supplementary materials so cannot comment on those.
Response: We can confirm that these were uploaded during original submission